# Lnk/Sh2b3 regulates initiation and severity of autoimmune insulitis and contributes to diabetes risk

Mari Tenno[1,2], Satoshi Takaki[1]

The Lnk/Sh2b3 adaptor protein functions as a regulatory molecule for cytokine signaling in lymphohematopoiesis. A missense variant of the *LNK/SH2B3* gene is reportedly a risk variant common to several autoimmune diseases, including type 1 diabetes (T1D). However, roles of Lnk in T1D development remain elusive. We found that *Lnk*$^{-/-}$ mice showed increased susceptibility to diabetes following treatment with fairly low doses of streptozotocin, manifested by hyperglycemia and insulitis accompanied by accumulation of CD8$^+$ T-cells and loss of pancreatic *β* cells. The high susceptibility of *Lnk*$^{-/-}$ mice to islet damage was abolished in crosses with *Rag2*$^{-/-}$ mice lacking lymphocytes or *MyD88*$^{-/-}$ mice carrying various defects in activation of innate immune cells. In *Lnk*$^{-/-}$ mice pancreata, dendritic cell (DC) fractions were altered and showed augmented expression of CD40 and IL-27. Treatment with anti-CD40L or anti-GM-CSF antibodies suppressed *β* cell damage and prevented diabetes. Thus, Lnk regulates T-cell priming and expansion via GM-CSF- and possibly IL-27-dependent activation of pancreatic DCs after islet damage.

## Introduction

Type 1 diabetes (T1D) is an autoimmune disease that results in the killing of pancreatic islet *β* cells and metabolic failure. Consequently, patients require lifelong insulin treatment (Ciecko et al, 2019; Herold et al, 2024). For T1D to develop, an immune response must first be elicited against *β* cell antigens. Second, destruction of *β* cells requires the establishment of a strong pro-inflammatory response. Third, regulatory control of the autoimmune reactions must be ineffective, thereby allowing the response to become chronic and to destroy the *β* cells. Considerable effort has been made to identify the immunological mechanisms causing *β* cell destruction. It is now known that many cell subsets participate in the pathogenesis. In recent-onset T1D patients who express the HLA-A2 (A*0201) allele, CD8$^+$ T-cells are found that recognize peptides generated from the preproinsulin signal peptide (Skowera et al, 2008). These T-cells are able to kill *β* cells in vitro via cytotoxic granule-mediated killing involving perforin and granzymes (Knight et al, 2013). To mount an effective CD8$^+$ T-cell response, dendritic cells (DCs) need to be licensed to cross-present exogenous antigens through MHC class I, which normally present endogenous peptides (Joffre et al, 2012). For DCs to acquire cross-presenting capacity, they must be activated, usually through interaction with activated CD4$^+$ T helper (Th) cells (Smith et al, 2004). Thus, the presence of islet-specific CD4$^+$ T-cells and their cytokine production is crucial for priming an immune response against the *β* cells. Depletion of CD4$^+$ T-cells in non-obese diabetic (NOD) mice leads to decreased incidence of disease and even reverses overt diabetes, emphasizing their importance for disease development (Makhlouf et al, 2004). DCs are also necessary for initiating the anti-islet immune response, and if they are eliminated, no disease develops (Saxena et al, 2007).

T1D occurs in individuals with a genetic predisposition. In such patients, environmental or immunological events trigger disease onset and progression (Herold et al, 2024). Genome-wide association studies (GWAS) have identified more than 50 loci significantly linked to T1D in humans (Todd et al, 2007; Cooper et al, 2008; Barrett et al, 2009; Bradfield et al, 2011; Evangelou et al, 2014; Fortune et al, 2015; Onengut-Gumuscu et al, 2015). Among them, a missense mutation (R262W) present within the *LNK/SH2B3* gene has been reported as a risk variant for several autoimmune diseases, including TID and celiac disease, as well as for cardiovascular disorders such as hypercholesterolemia, myocardial infarction, and hypertension (Todd et al, 2007; Hunt et al, 2008; Smyth et al, 2008; Coenen et al, 2009).

Lnk/Sh2b3 (Lnk) is an adapter protein that negatively regulates cytokine signaling in lymphohematopoiesis (Takaki et al, 2000, 2002; Gery & Koeffler, 2013; Katayama et al, 2014; Mori et al, 2014). Lnk is expressed in hematopoietic cells and is implicated in the integration and regulation of multiple signaling events. *Lnk*$^{-/-}$ mice overproduce B-cells (Takaki et al, 2000) and hematopoietic stem cells (Takaki et al, 2002). We showed that Lnk prevents the pathogenic expansion of CD8$^+$ T-cells, leading to damage of intestinal tissue by IL-15 (Katayama et al, 2014). Lnk also

[1]Department of Immune Regulation, The Research Center for Hepatitis and Immunology, National Institute of Global Health and Medicine, Japan Institute for Health Security, Chiba, Japan   [2]Division of Cancer Cell Biology, Research Institute for Biomedical Sciences (RIBS), Tokyo University of Science, Chiba, Japan

Correspondence: lbstakaki@hospk.ncgm.go.jp, takaki.s@jihs.go.jp

regulates the ability of DCs to support Th1 or Treg cells in response to GM-CSF and IL-15 (Mori et al, 2014). With regard to the regulation of glucose metabolism, we have previously shown that *Lnk*-deficiency increases the IL-15-reactivity of type 1 innate lymphocytes in adipose tissue. That alteration results in adipose inflammation due to overproduction of INF-γ. Such inflammation causes glucose intolerance and insulin resistance (Mori et al, 2018). However, the relationship between the function of Lnk and pathological conditions in T1D development has yet to be fully elucidated.

In this study, we sought to reveal Lnk functions as they relate to the development of insulin insufficiency and the pathogenesis of diabetes. We found that the loss of Lnk exacerbated pancreatic islet inflammation and impaired glucose tolerance. We show that the predominant effect of Lnk on islet β cell destruction occurs through the hematopoietic compartment. The Lnk-dependent regulatory system functions in the process of lymphocyte priming via GM-CSF-dependent activation of pancreatic DCs after initial β cell destruction. Loss or reduced functions of Lnk leads to enhanced activation of pancreatic DCs, which leads to exacerbated autoimmune islet damage and increased risk for T1D development.

## Results

### Loss of Lnk accelerates diabetes induced by streptozotocin (STZ)

When $Lnk^{-/-}$ mice were fed a normal chow diet, blood glucose levels were slightly elevated (Fig S1A) due to adipose inflammation as previously reported (Mori et al, 2018); however, $Lnk^{-/-}$ mice in steady state did not show any signs of insulitis on histological or flow cytometric analysis (Fig S1B–F). To investigate how Lnk functions affect the susceptibility to diabetes, we used STZ, a selective cytotoxic agent against insulin-generating pancreatic β cells, resulting in β cell apoptosis and autoimmune diabetes (Paik et al, 1980). We administrated 50 mg/kg STZ on three consecutive days. Destruction of β cells was relatively mild in this condition, and hyperglycemia was observed in $Lnk^{-/-}$ but not in WT mice (Fig 1A). Histological analysis of the pancreas revealed loss of insulin-producing β cells in STZ-injected $Lnk^{-/-}$ but not in WT mice (Fig 1B and C). Although untreated $Lnk^{-/-}$ mice did not show any signs of insulitis in histological analysis (Fig S1B), low-dose STZ treatment induced severe cell infiltration into islets of $Lnk^{-/-}$ mice compared with WT mice (Fig 1B and C). Only mild damage to β cells resulted in uncontrolled immune cell infiltration into islets, leading to β cell loss in the absence of Lnk.

### $Lnk^{-/-}$ T-cells and DCs are responsible for the loss of β cells in islets

We next used flow cytometry to analyze inflammatory cells that had infiltrated into the pancreas. We found that CD45$^+$ hematopoietic cells were markedly increased in STZ-injected $Lnk^{-/-}$ mice (Fig 1D). Among the CD45$^+$ cells, the frequency and absolute number of CD8$^+$ cells were increased in $Lnk^{-/-}$ mice 7 d after STZ treatment (Fig 1E). The numbers of B220$^+$, CD11b$^+$F4/80$^-$, and

CD11b$^+$F4/80$^+$ cells were not significantly altered (Fig S2A). For conventional DC (cDC) compartment, we gated on CD45$^+$F4/80$^-$CD64$^-$CD11c$^+$MHCII$^+$ to exclude cells derived from macrophages and monocytes, and further divided into CD103$^+$ cDC1 and CD11b$^+$ cDC2 (Fig 1F). Since there were no significant changes in CD11b$^+$ cDC2, we focused on CD103$^+$ cDC1 in further analysis. In steady state, the frequency of CD8$^+$ cells and cDC1 in the pancreas (Fig S1C and D) and pancreatic lymph nodes (Fig S1E and F) were comparable between untreated $Lnk^{-/-}$ and WT mice, implying that augmented cell infiltration in $Lnk^{-/-}$ pancreas was induced in response to the mild injury of β cells.

To examine the contribution of T-cells, we crossed $Lnk^{-/-}$ mice with $Rag2^{-/-}$ mice lacking T- and B-cells. $Lnk^{-/-}Rag2^{-/-}$ mice did not develop STZ-induced diabetes and did not lose pancreatic β cells (Fig S3A). Those data indicated that the augmented loss of β cells in the $Lnk^{-/-}$ pancreata was promoted by cytotoxic injury of β cells mediated by T-cells. To further confirm the identity of cells responsible for exacerbated diabetes in $Lnk^{-/-}$ mice, we employed bone marrow (BM) chimeric mice (Fig S4). WT recipients transplanted with $Lnk^{-/-}$ BM cells, but not with WT BM cells, completely reproduced the hyperglycemia and insulitis seen in $Lnk^{-/-}$ mice and developed diabetes after low-dose of STZ injections (Figs 1 and S4A). $Lnk^{-/-}$ recipients transplanted with $Lnk^{-/-}$ BM cells, but not with WT BM cells, also developed diabetes after STZ injection (Fig S4B). These data suggest that $Lnk^{-/-}$ hematopoietic cells, but not radio-resistant non-hematopoietic cells, are responsible for β cell destruction induced by low-dose STZ treatment. We created $MyD88^{-/-}Lnk^{-/-}$ mice and found that $MyD88^{-/-}Lnk^{-/-}$ as well as WT or $MyD88^{-/-}$ mice did not develop diabetes after low-dose STZ injection (Fig S3B). Furthermore, WT recipients transplanted with $MyD88^{-/-}Lnk^{-/-}$ BM cells did not develop diabetes in contrast to those transplanted with $Lnk^{-/-}$ BM cells (Fig S4A). Thus, MyD88-mediated responses in radio-sensitive cells were required to activate and accumulate CD8$^+$ T-cells in pancreatic islets, leading to the loss of β cells in $Lnk^{-/-}$ mice.

### Signaling is affected by the absence of Lnk in pancreatic DCs and T-cells

MyD88 is essential for signaling through all TLRs except TLR3 (Kawai & Akira, 2010), and it is important for the activation and maturation of DCs, namely induction of various costimulatory molecules such as CD40, CD80, etc. (Akira et al, 2001). Mononuclear cells that have infiltrated islets express TLR2, 3 and 4 in humans (Devaraj et al, 2008; Aida et al, 2011). cDC1 plays essential roles in CD8$^+$ T-cell activation as antigen-presenting cells (APCs). Next, we analyzed cDC1 in the pancreas and found that the expression of TLR4, but not TLR2 or TLR3, was significantly up-regulated in STZ-injected $Lnk^{-/-}$ mice (Fig 2A). CD40 and CD80 expressions on $Lnk^{-/-}$ cDC1 were markedly enhanced after STZ administration (Fig 2B). CD86 levels on $Lnk^{-/-}$ and WT cDC1 were comparable (Fig 2B) as seen in NOD mice (Price et al, 2014), indicating that $Lnk^{-/-}$ cDC1 with high CD40 and CD80 expression did not result from overall activation of DCs, but was restricted to specific pathways, including CD40 as observed in NOD mice (Price et al, 2014). The expression of CD40 on other APCs, namely B220$^+$ or CD11b$^+$F4/80$^+$ cells, remained unchanged (Fig S2B).

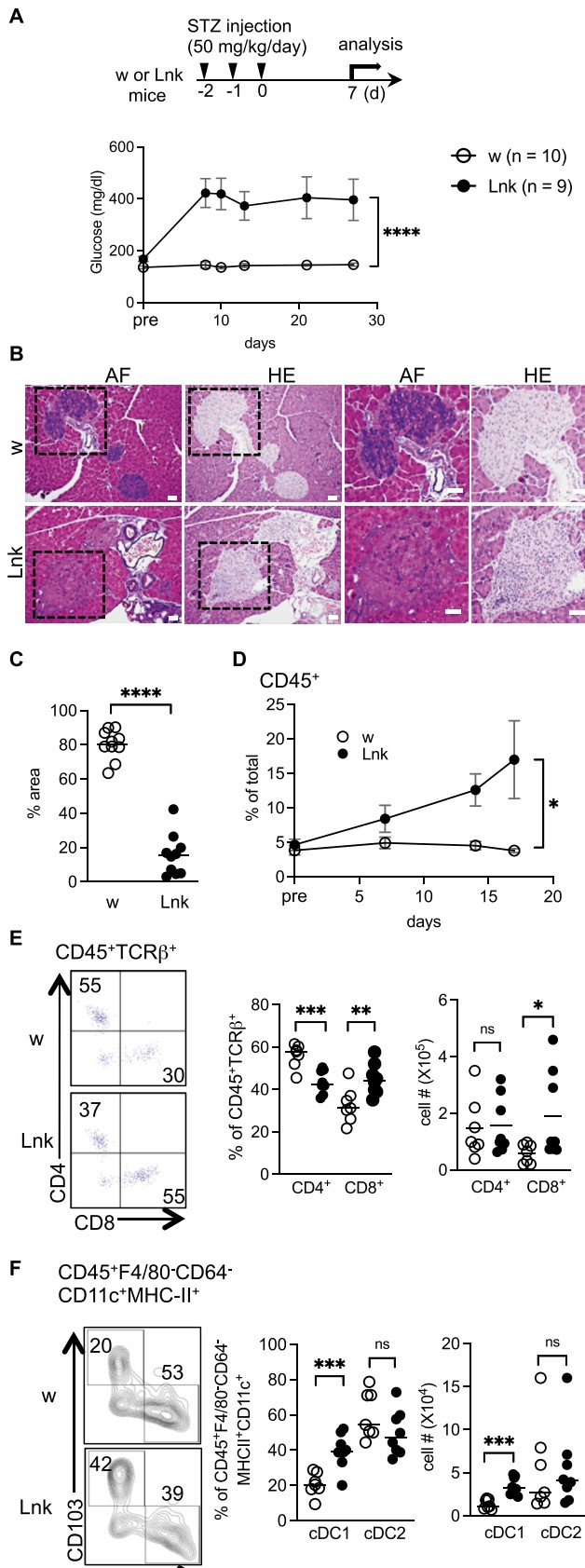

The combined stimulation of DCs by TLR and CD40 induced CD70 expression and IL-27 production in vivo, leading to a massive expansion of CD8[+] T-cells (Bullock & Yagita, 2005; Sanchez et al, 2007; Pennock et al, 2014; Meka et al, 2015). GWAS identified IL-27 as one of the genes related to T1D pathogenesis (Wang et al, 2008; Barrett et al, 2009). We found that $Lnk^{-/-}$ cDC1 up-regulated CD70 (Fig 2B) and produced more IL-27 after STZ treatment than did WT cells (Fig 2C). Following STZ treatment, IL-27Rα expression on $Lnk^{-/-}$ cDC1 was not significantly altered after 7 d, but it did increase after 14 d (Fig 2C), suggesting IL-27 worked in an autocrine manner on pancreatic cDC1. Phosphorylation of STAT3 markedly increased in $Lnk^{-/-}$ cDC1, whereas STAT1 phosphorylation was unchanged (Fig 2D). Phosphorylation of STAT5 in $Lnk^{-/-}$ cDC1 also markedly increased (Fig 2D, discussed below). $Lnk^{-/-}$ cDC1 also showed high expression of programmed death-ligand 1 (PD-L1) (Fig 2E). However, they did not seem to have a typical suppressive phenotype as the expression of other co-inhibitory molecules such as PD-L2 and PD-1 was low (Fig 2E). $Lnk^{-/-}$ CD8[+] T-cells showed elevated IL-27Rα, pSTAT3, and pSTAT1 expression, especially in the pancreas (Fig 3A and B). IL-27 exerts its effects through STAT1/3-dependent cascades (Hunter & Kastelein, 2012; Pennock et al, 2014; Morita et al, 2021). The frequency of Sca-1[+], CD122[+], or T-box expressed in T cells (T-bet)[+] cells significantly increased among $Lnk^{-/-}$ CD8[+] T-cells compared with WT CD8[+] T-cells (Fig 3C–E). Thus, IL-27-dependent signaling promotes differentiation and accumulation of activated CD8[+] effector T-cells in the pancreatic islets without Lnk.

## Prevention of β cell destruction by blocking DC activation or DC/T-cell interaction

Early treatment of NOD mice with anti-CD40L prevents insulitis and diabetes (Balasa et al, 1997). $Lnk^{-/-}$ cDC1 showed increased

**Figure 1. Lnk-deficient mice showed exacerbated diabetic conditions in response to low-dose STZ injection.**
**(A)** The top panel shows the experimental setup for STZ administration. Blood glucose levels were measured in 8- to 12-wk-old WT (open circles) or $Lnk^{-/-}$ mice (filled circles) before the first injection (pre) and after the final injection of low-dose STZ from days 7 to 29. The statistical comparison was performed using two-way ANOVA with Sidák correction. ****$P$ < 0.0001. **(B)** Representative sections obtained from the pancreata of STZ-treated WT (upper panels) or $Lnk^{-/-}$ (lower panels) mice. Sections were stained with aldehyde-fuchsin (AF) or hematoxylin and eosin (HE). Tissue was examined by light microscopy (original magnification 200×). Boxes indicated by dashed lines were further magnified and presented in the right four panels. Scale bars indicate 40 μm. **(C)** The graph showed the percentage of the purple area (β cells) in the islets. **(D, E, F)** Flow cytometric analysis of single cell suspensions were prepared by collagenase-digestion of the pancreata obtained from STZ-injected mice. **(D)** Proportions of CD45[+] cells in total cells analyzed at 7, 14, and 17 d after STZ-treatment (n = 9–10 mice per group). **(E)** Representative dot plots show CD4 and CD8 expression, which were gated on CD45[+]TCRβ[+] cells (left). Proportions of CD4[+] and CD8[+] cells in CD45[+]TCRβ[+] cells (middle), and the numbers of CD4[+] and CD8[+] cells infiltrated into the pancreata (right). **(F)** Representative contour plots show CD11b and CD103 expression, which were gated on CD45[+]F4/80[−]CD64[−]CD11c[+]MHC-II[+] cells (left). Proportions of cDC1 (CD11b[−]CD103[+]) and cDC2 (CD11b[+]CD103[−]) cells in CD45[+]F4/80[−]CD64[−]CD11c[+]MHC-II[+] cells (middle), and the numbers of cDC1 and cDC2 cells infiltrated into the pancreata (right). Error bars show SD. *$P$ < 0.05; **$P$ < 0.01; ***$P$ < 0.001, ****$P$ < 0.0001, ns, not significant (Student's unpaired $t$ test).

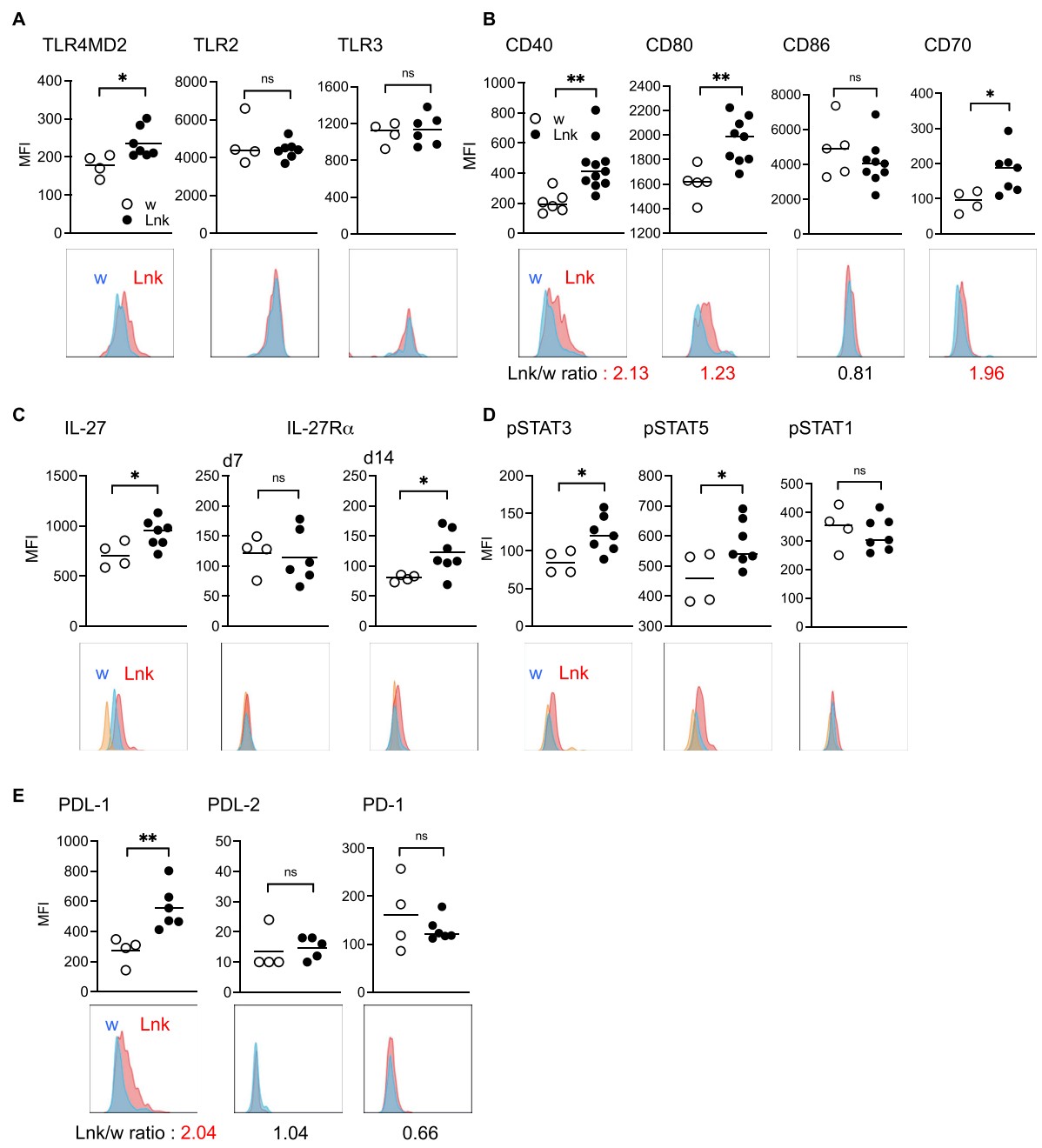

**Figure 2. Enhanced activation of dendritic cells in the pancreata of *Lnk*-deficient mice treated with low-dose STZ.**
**(A, B)** Histograms show the expression of TLR4MD2, TLR2, and TLR3 (A), CD40, CD80, CD86, and CD70 (B), in cDC1 in the pancreata from WT (blue) or *Lnk*$^{-/-}$ (red) mice 7 d after STZ administration (lower). Graphs summarize the mean fluorescence intensity (MFI) of indicated molecules from WT (open circles) or *Lnk*$^{-/-}$ (closed circles) mice (upper). Each dot represents an individual mouse. Lnk/w ratio in MFI is shown below each histogram. **(C)** Histograms showing the expression IL-27 (day 7) and of IL-27Rα on cDC1 in the pancreata from WT (blue) or *Lnk*$^{-/-}$ (red) mice 7 d (lower middle) and 14 d (lower right) after STZ administration. Orange shows fluorescence minus one control (FMO) controls. Graphs summarize the MFI of IL-27 and of IL-27Rα from WT (open circles) or *Lnk*$^{-/-}$ (closed circles) mice (upper). Each dot represents an individual mouse. **(D, E)** Histograms show the expression of pSTAT3, pSTAT5. pSTAT1 (D), PDL-1, PDL-2, and PD-1(E) in cDC1 in the pancreata from WT (blue) or *Lnk*$^{-/-}$ (red) mice 7 d after STZ administration (lower). Orange shows fluorescence minus one control (FMO) controls. Graphs summarize the MFI of indicated molecules from WT (open circles) or *Lnk*$^{-/-}$ (closed circles) mice (upper). Each dot represents an individual mouse. Lnk/w ratio in MFI is shown below each histogram. *$P < 0.05$; **$P < 0.01$. ns, not significant (Student's unpaired *t* test).

expression of CD40 (Fig 2B), whereas other APCs did not (Fig S2B). Therefore, we asked whether anti-CD40L could block the induction of diabetes in *Lnk*$^{-/-}$ mice. Anti-CD40L-treated *Lnk*$^{-/-}$ mice

manifested no insulitis after STZ injection, whereas non-treated control mice showed severe insulitis (Fig 4A–D). The activation of cDC1, as demonstrated by expressions of CD40 and CD80, was not

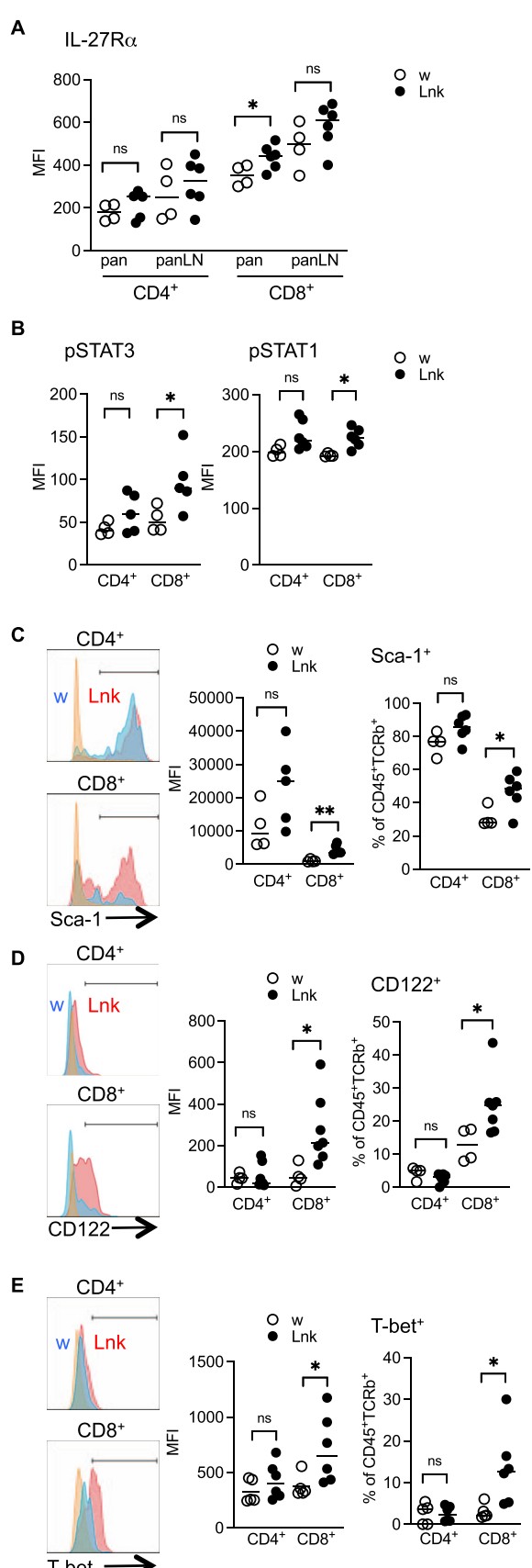

**Figure 3. Augmented infiltration and activation of T cells in the pancreata in** *Lnk*-deficient mice treated with low-dose STZ.

**(A)** Graphs show a summary of the MFI values of IL-27Rα expression on CD4+ or CD8+ T cells in the pancreata or peri-pancreatic lymph nodes prepared from WT (open circles) or *Lnk*−/− (closed circles) mice 14 d after STZ administration. **(B)** Graphs show summaries of the MFI of pSTAT3 and pSTAT1 in CD4+ or CD8+ T cells in the pancreata from WT (open circles) or *Lnk*−/− (closed circles) mice 7 d after STZ administration. Each dot represents an individual mouse. **(C, D)** Histograms show the expression of Sca-1 (C) and CD122 (D) on CD4+ or CD8+ T cells in pancreata from WT (blue) or *Lnk*−/− (red) mice 7 d after STZ administration. Orange shows unstained controls. **(C, D)** Graphs showing summary of MFI and the proportions of Sca-1+ (C) or CD122+ (D) cells on CD4+ or CD8+ T cells from WT (open circles) or *Lnk*−/− (closed circles) mice. Each dot represents an individual mouse. **(E)** Histograms showing the expression of T-bet in CD4+ or CD8+ T cells in the pancreata from WT (blue) or *Lnk*−/− (red) mice 7 d after STZ administration. Orange shows fluorescence minus one (FMO) controls. Graphs show a summary of the MFI and the proportions of T-bet+ cells on CD4+ or CD8+ T cells from WT (open circles) or *Lnk*−/− (closed circles) mice day 7 after STZ administration. Each dot represents an individual mouse. *P < 0.05. ns, not significant (Student's unpaired t test).

altered and remained high (Fig 4D). B cell depletion by anti-CD20 administration did not prevent the increase of blood glucose level in STZ-treated *Lnk*−/− mice (Fig S5A and B). Those data implied that anti-CD40L-treatment blocked DC/T-cell interaction via CD40‑CD40L.

Pancreatic cDC1 showed enhanced phosphorylation of STAT5 in STZ-treated *Lnk*−/− mice (Fig 2D). We previously reported that growth responses of BM-derived DCs to GM-CSF signals through phospho-STAT5 were enhanced (Mori et al, 2014). GM-CSF is important for the induction of CD8+ T-cell immunity through the regulation of nonlymphoid tissue DC homeostasis (Greter et al, 2012), and is a critical player in several autoimmune diseases (Achuthan et al, 2021; Ingelfinger et al, 2021). Therefore, we examined anti-GM-CSF administration and found that the GM-CSF neutralization protected *Lnk*−/− mice from STZ-induced severe insulitis resulting in diabetes (Fig 5A–C). GM-CSF neutralization blocked the up-regulation of CD40 and PDL-1 as well as IL-27 and IL-27Rα in *Lnk*−/− cDC1 in the pancreas (Fig 5D).

## Discussion

We found that *Lnk*−/− mice showed increased susceptibility to loss of pancreatic β cells following treatment with fairly low doses of streptozotocin. *Lnk*−/− mice manifested hyperglycemia and insulitis accompanied by accumulation of CD8+ T-cells in response to the mild injury of β cells. The high susceptibility of *Lnk*−/− mice to islet damage was abolished in crosses with *Rag2*−/− or *MyD88*−/− mice. We found that *Lnk*−/− T-cells and DCs were responsible for the loss of β cells in islets and signaling in pancreatic DCs and T-cells was affected by the absence of Lnk.

In NOD mice, diabetes hardly progresses if cDC1 is ablated by *Batf3*-deficiency (Ferris et al, 2014). In addition, myeloid cell-specific *MyD88*-deficiency considerably protected mice from developing STZ-induced T1D and delayed the onset of diabetes in NOD mice (Androulidaki et al, 2018). In *Lnk*−/− mice pancreata,

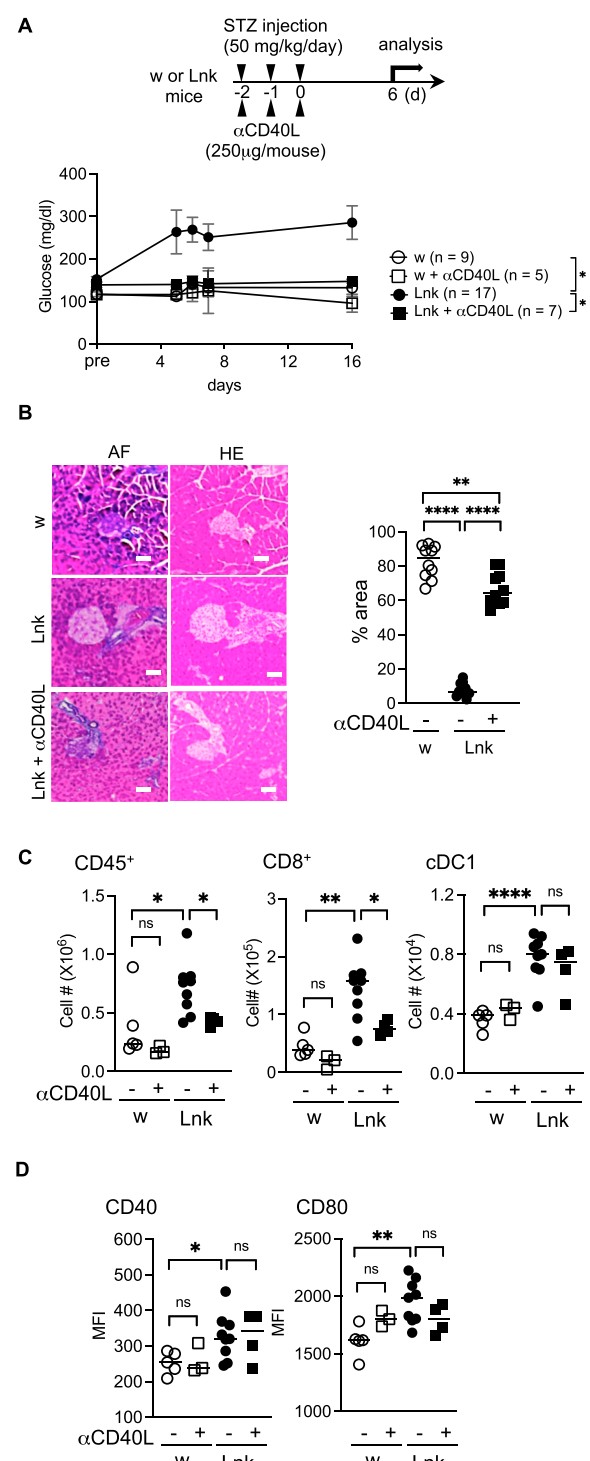

**Figure 4. Administration of anti-CD40L antibody reduced the onset of diabetes mellitus in *Lnk*-deficient mice.**
**(A)** The top panel shows a schematic diagram for anti-CD40L administration during MLD-STZ. Blood glucose levels were measured in WT (open circles), WT with anti-CD40L (open squares), $Lnk^{-/-}$ (filled circles), or $Lnk^{-/-}$ with anti-CD40L (closed squares) mice before the first injection (pre) and after the final injection of STZ from days 6 to 16. The statistical comparison was performed using two-way ANOVA with Šidák correction. *$P < 0.05$. **(B)** Representative sections obtained from the pancreata of STZ-treated WT (upper panels), $Lnk^{-/-}$ (middle panels), or $Lnk^{-/-}$ with anti-CD40L (lower panels) mice. Sections were stained

cDC1 fraction was altered and showed augmented expression of CD40 and IL-27 (Figs 1F and 2B and C). The roles of IL-27 in T1D are still controversial, and the results are conflicting. Whereas IL-27 induces massive mononuclear cell infiltration after STZ treatment (Mensah-Brown et al, 2006), mice lacking IL-27 or IL-27Rα manifest increased blood sugar levels and islet infiltration after STZ-injection (Fujimoto et al, 2011). On the other hand, NOD mice lacking IL-27 or IL-27Rα completely resist diabetes development (Ciecko et al, 2019). Activated DCs produce IL-27, and blocking IL-27 by antibodies significantly delays the onset of T1D after the transfer of splenocytes into lymphocyte-deficient NOD-Scid recipients (Wang et al, 2008). IL-27 is a unique cytokine with reported immunostimulatory and immunosuppressive effects that are exerted through STAT1/3-dependent cascades (Hunter & Kastelein, 2012; Pennock et al, 2014; Morita et al, 2021) and thought to be involved in various autoimmune diseases including T1D (Meka et al, 2015). Expression of IL-27Rα on myeloid DCs from T1D patients is increased, and the resulting IL-27 signaling enhances STAT3 phosphorylation and the expression of PD-L1 (Carl & Bai, 2007; Parackova et al, 2020). Upon TLR agonist stimulation, STAT3 is rapidly recruited to the PD-L1 promoter in DCs (Wölfle et al, 2011). These findings prompted us to further study cell-based mechanisms leading to T1D, and we focused on Lnk and IL-27 functions.

The ratio between co-inhibitory and co-stimulatory molecules critically determines the functionality of APCs (Tokita et al, 2008; Shen et al, 2010). In many cases, PDL-1-expressing inhibitory cells show decreased expression of activation markers such as CD40 and CD80. On the other hand, it has recently been demonstrated that PDL-1 does not act as an inhibitory molecule when CD80 is expressed on the same cell (Sugiura et al, 2019; Zhao et al, 2019). If co-expressed on the same APC, the interaction of PD-L1 and CD80 takes place in cis (on the same cell), preventing PD-1 from engaging with PD-L1 in trans (across cell membranes) and not affecting CD80 binding with CD28 (Sugiura et al, 2019; Zhao et al, 2019). In the pancreas of $Lnk^{-/-}$ mice, cDC1 showed elevated expression of PDL-1, but not of other inhibitory molecules such as PDL-2 and PD-1 (Fig 2E). They also showed increased expression of CD80, which could block inhibitory functions of PDL-1 in cis, and high expression of other stimulatory molecules such as CD40 and CD70 (Fig 2B). These findings suggest that cDC1 manifests increased stimulatory ability without Lnk.

Given the $Lnk^{-/-}$ cDC1 phenotypes with enhanced expression of both CD40 and CD80, we analyzed T-cells with regard to IL-27 stimulation and Th1 differentiation. $Lnk^{-/-}$ CD8$^+$ T-cells showed

with aldehyde-fuchsin (AF) or hematoxylin and eosin (HE). Tissue was examined by light microscopy (original magnification 200×). Scale bars indicate 40 $\mu$m. The graph showed the percentage of the purple area ($\beta$ cells) in the islets (right). **(C)** The numbers of CD45$^+$ cells in total cells (left), the numbers of CD8$^+$ cells in CD45$^+$TCR$\beta^+$ cells (middle), or the numbers of cDC1 in CD45$^+$F4/80$^-$CD64$^-$CD11c$^+$MHC-II$^+$ cells (right) infiltrated in the pancreata from WT (open circles), WT with anti-CD40L (open squares), $Lnk^{-/-}$ (closed circles) or $Lnk^{-/-}$ with anti-CD40L (closed squares) mice. **(D)** The expression of CD40 (left) or CD80 (right) on cDC1 in the pancreata from WT (open circles), WT with anti-CD40L (open squares), $Lnk^{-/-}$ (closed circles) or $Lnk^{-/-}$ with anti-CD40L (closed squares) mice. *$P < 0.05$; **$P < 0.01$; ***$P < 0.001$. ns, not significant (Student's unpaired t test).

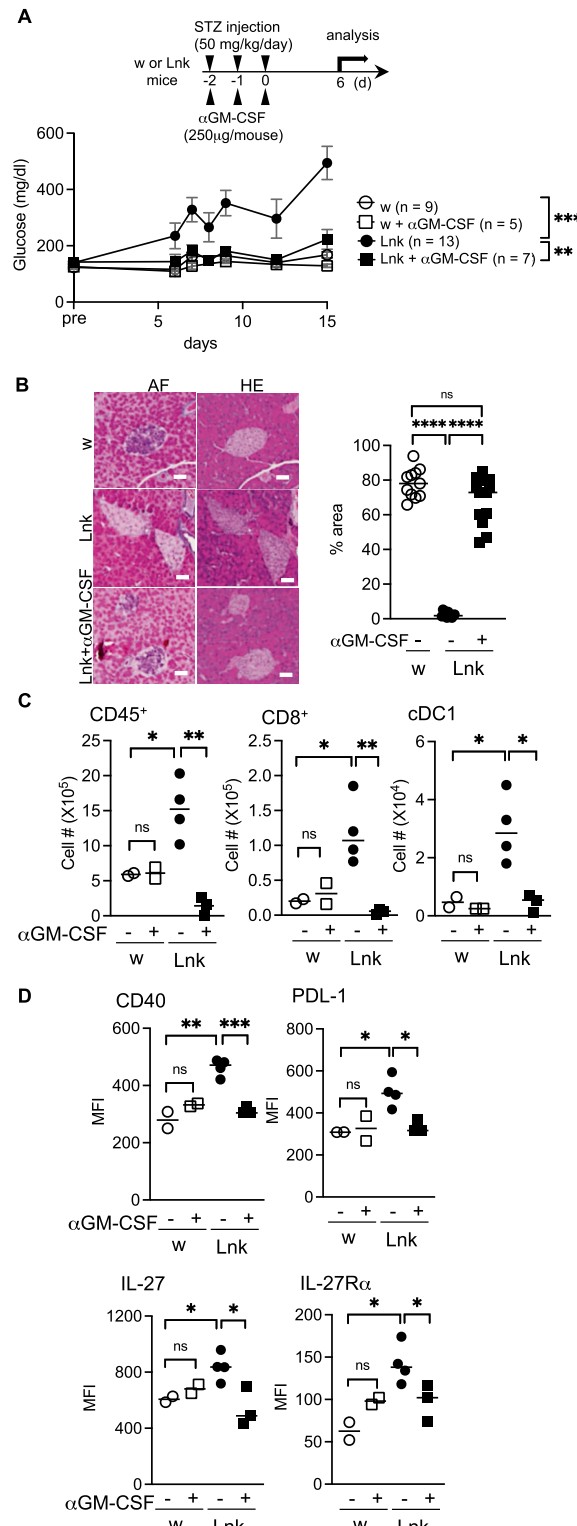

**Figure 5. Administration of anti-GM-CSF antibody reduced the activation of dendritic cells in the pancreata and prevented the onset of diabetes mellitus.**

**(A)** The top panel shows a schematic diagram for anti-GM-CSF administration during MLD-STZ. Blood glucose levels were measured in WT (open circles), WT with anti-GM-CSF (open squares), $Lnk^{-/-}$ (filled circles), or $Lnk^{-/-}$ with anti-GM-CSF (closed squares) mice before the first injection (pre) and after the final

elevated IL-27Rα expression in the pancreas, and Sca-1[+], CD122[+], or T-bet[+] cells was increased among $Lnk^{-/-}$ CD8[+] T-cells. IL-27Rα expression is highly up-regulated during effector T-cell differentiation induced by naïve T-cells (Carl & Bai, 2007). IL-27 increased CD122 expression via STAT3 (Morishima et al, 2005) and Eomes to support the proliferation and survival of T-cells by CD122-mediated IL-2/15 signaling (Intlekofer et al, 2005). IL-27 also induces T-bet, a master transcriptional regulator for Th1 differentiation (Szabo et al, 2000), and Sca-1 in effector T-cells (Liu et al, 2017). Thus, it seems that IL-27-dependent signaling contributes in part to the accumulation of CD8[+] effector T-cells in the pancreas without Lnk. Furthermore, experiments, however, will be required to directly confirm if IL-27 plays critical roles in developing diabetes after β cell injury in the absence of Lnk.

$MyD88^{-/-}Lnk^{-/-}$ mice did not develop diabetes after low-dose STZ injection. Treatment of $Lnk^{-/-}$ mice with anti-CD40L prevents insulitis and diabetes after STZ injection. In addition to DCs, B cells could also be activated by MyD88-mediated signals and could expand T cells via CD40–CD40L interaction in the process for developing diabetes. We examined the involvement of B cells by depleting B cells using anti-CD20. Repeated injection of anti-CD20 at d0, d2, and d5 efficiently depleted B cells in $Lnk^{-/-}$ or WT type mice within 7 d (Fig S5B). Anti-CD20-treated $Lnk^{-/-}$ mice showed higher blood glucose level after STZ injection, comparable to treated with control IgG antibodies (Fig S5A). WT mice did not show increase of blood glucose after anti-CD20-treatment. GM-CSF neutralization, which has little effect on B cells because of the expression of GM-CSFR α chain specific for GM-CSF is low in B-cells (Achuthan et al, 2021), protected $Lnk^{-/-}$ mice from insulitis and diabetes induced by low-dose STZ injection. These data further implied that anti-CD40L-treatment blocked DC/T-cell interaction, but not B/T-cell interaction via CD40–CD40L.

Our previous study revealed that DC numbers are increased in the spleen and lymph nodes in $Lnk^{-/-}$ mice. Moreover, growth responses of BM-derived DCs to GM-CSF signals through phospho-STAT5 were augmented (Mori et al, 2014). In addition, mature DCs obtained from $Lnk^{-/-}$ spleens are hypersensitive and showed enhanced responses to IL-15 and GM-CSF (Mori et al, 2014). GM-CSF is important for the induction of CD8[+] T-cell immunity through the regulation of nonlymphoid tissue DC homeostasis (Greter et al, 2012) and is a critical player in several autoimmune diseases (Achuthan et al, 2021; Ingelfinger et al, 2021). In the pancreas, GM-

injection of low-dose STZ from days 6 to 15. The statistical comparison was performed using two-way ANOVA with Šidák correction. **P < 0.01; ***P < 0.001. **(B)** Representative sections obtained from the pancreata of STZ-treated WT (upper panels), $Lnk^{-/-}$ (middle panels), or $Lnk^{-/-}$ treated with anti-GM-CSF (lower panels) mice. Sections were stained with aldehyde-fuchsin (AF) or hematoxylin and eosin (HE). Tissue was examined by light microscopy (original magnification 200×). Scale bars indicate 40 $\mu$m. The graph showed the percentage of the purple area (β cells) in the islets (right). **(C)** The numbers of CD45[+] cells in total cells, CD8[+] cells in CD45[+]TCRβ[+] cells, and cDC1 in CD45[+]F4/80[−]CD64[−]CD11c[+]MHC-II[+] cells infiltrated into the pancreata from WT (open circles), WT with anti-GM-CSF (open squares), $Lnk^{-/-}$ (closed circles) or $Lnk^{-/-}$ with anti-GM-CSF (closed squares) mice were plotted. **(D)** The expression of CD40, PDL-1, IL-27, or IL-27Rα on cDC1 in the pancreata from WT (open circles), WT with anti-GM-CSF (open squares), $Lnk^{-/-}$ (closed circles) or $Lnk^{-/-}$ with anti-GM-CSF (closed squares) mice. *P < 0.05; **P < 0.01; ***P < 0.001. ns, not significant (Student's unpaired t test).

CSF could be a critical initiator of enhanced activation of *Lnk*-deficient cDC1. Once activated, further activation of cDC1 and CD8[+] effector T-cells could result from IL-27/IL-27R loops.

Recently, Watson et al have demonstrated an important T-cell intrinsic regulatory role for Lnk in T1D (Watson et al, 2025). *Lnk*-deficiency exacerbated diabetes in multiple mouse models, including NOD. *Lnk*-deficiency potently promotes pathogenic CD8[+] T cells with enhanced survival and effector function by enhanced activation of the JAK/STAT pathways via IL-2, IL-15, and IFNγ (Watson et al, 2025). JAK inhibitors have shown efficacy in preserving β cell function in the NOD mouse model (Trivedi et al, 2017) and in human T1D subjects (Waibel et al, 2023). Our findings support regulatory functions of Lnk in response to β cell injury in the STZ model of autoimmune diabetes. We revealed enhanced activation of cDC1 and expansion of CD8[+] T cells in the absence of Lnk, potentially mediated through GM-CSF and IL-27, both of which also activate JAK/STAT pathways. Overall, these data provide further support for the use of JAK inhibitors in T1D, accompanied by reduced functions of Lnk with the risk variants of *LNK/SH2B3* gene (Wang et al, 2016) for T1D.

In conclusion, we identified Lnk as an important regulator preventing uncontrolled pancreatic tissue destruction leading to T1D. This was attributed to the enhanced activation of cDC1 by GM-CSF, possibly followed by autocrine stimulation with IL-27/IL-27R. Activation of cDC1 leads to pronounced induction and activation of CD8[+] T-cells via IL-27/IL-27R and finally to loss of β cells in the pancreas. The results advance our understanding of the functions of a T1D-related gene and also suggest potential approaches for preventive therapy for patients in high-risk factor groups for autoimmune diseases.

# Materials and Methods

### Mice

C57BL/6 mice were purchased from CLEA Japan. C57BL/6 mice congenic for the Ly5 locus (CD45.1), $Rag2^{-/-}$ and $Lnk^{-/-}$ (Takaki et al, 2000) were previously described. $MyD88^{-/-}$ (Adachi et al, 1998) were provided from OrientalBioService. Mice were maintained under specific pathogen-free conditions and used for experiments at 6–14 wk of age. This study was approved by the Animal Care and Use Committee of the Research Institute, National Center for Global Health and Medicine (approval No. 2024-A014). All mice were handled in accordance with the Guidelines for Animal Experiments of the Research Institute, National Center for Global Health and Medicine and the ARRIVE (Animal Research: Reporting of In Vivo Experiments) guidelines. Sevoflurane was used as an inhalation anesthetic to anesthetize mice.

### Induction of diabetes by multiple low doses of STZ (MLD-STD)

Mice were given daily intraperitoneal injections of streptozotocin (50 mg/kg; Sigma-Aldrich) for three consecutive days. Glucose concentrations were measured in venous blood using an automated glucometer.

### Cell preparation

Single cell suspensions from mouse pancreata were prepared as previously described (Tenno et al, 2020) with some modifications. Briefly, pancreatic tissues were chopped into small pieces and then digested with 0.2 mg/ml collagenase IV (Sigma-Aldrich) in RPMI supplemented with 2% FBS at 37°C for 30 min with shaking. To homogenize samples, we placed tissues in a syringe with an 18G needle and flushed them out 10 times and then filtered the suspension with a 70 $\mu$m cell strainer. Cells were collected by centrifugation at 280$g$ at 4°C for 5 min. The pellets were washed once with PBS and were subjected to analysis.

### Flow cytometry

Single cell suspensions were stained with anti-CD16/32 mAb (2.4G2; BD Biosciences) to prevent nonspecific binding of antibodies via FcR interactions. Then, cells were stained with the following antibodies purchased from BD Bioscience or eBioscience: CD3 (145-2C11), CD4 (RM4-5), CD8 (53-6.7), CD11b (M1/70), CD11c (HL3), CD40 (1C10), CD45 (30-F11), CD45.1 (A20), CD45.2 (104), CD64 (X54-5/7.1), CD80 (16-10A1), CD86 (GL1), CD103 (2E7), CD122 (TM-b1), F4/80 (BM8), IL-27 (MM27-7B1), IL-27Rα (W16125D), PD-1 (RMP1-30), PDL-1 (10F9G2), PDL-2 (TY25), pSTAT1 (4a), pSTAT3 (4/P-STAT3), pSTAT5 (47/Stat5), Sca-1 (D7), TCRβ (H57-597), TLR2 (QA16A01), TLR3 (11F8), TLR4 (SA15-21), TLR4MD2 (MTS510), MHC-II (M5/114.15.2), T-bet (4B10), and FoxP3 (FJK). For intracellular staining with antibodies, cells were permeabilized with eBioscience Foxp3/Transcription Factor Staining Buffer Set (00-5523-00). Flow cytometric analysis was performed with a FACSCanto II (BD Biosciences), and the data were analyzed using FlowJo (Tree Star) software.

### Histology

Pancreatic tissues were fixed in 4% PFA in PBS overnight, embedded in paraffin, and sectioned at a 5 $\mu$m thickness. The sections were stained with hematoxylin and eosin (MUTO PURE CHEMICALS) or aldehyde-fuchsin (MUTO PURE CHEMICALS). Tissue was examined by light microscopy (original magnification 200×).

### Bone marrow transplantation

Femoral bone marrow (BM) cells from donor mice were washed with PBS and counted. BM cells (5 × 10[6] per recipient mouse) were intravenously transfused into lethally irradiated (9 Gy) recipient (CD45.1 or CD45.2) mice. Three months later, PBLs were collected from the recipient mice, and chimerism was confirmed by their expression of CD45.1 or CD45.2.

### Treatment with antibodies in vivo

In blocking experiments, we intraperitoneally administered 250 $\mu$g of anti-CD40L antibody (MR-1; Bio X-cell) or anti-GM-CSF antibody (MP1-22E9; Bio X-cell), concurrently with STZ administration. For B cell depletion anti-CD20 (Anti-mCD20-mIgG2a InvivoFit, 10 mg/g BW; Invivogen) were intraperitoneally administered at -d2, d0, and d3.

## Statistical analysis

All statistical analyses were performed using Prism 6 software (GraphPad Software). Two groups were compared using the *t* test. Multiple groups were compared using one-way or two-way ANOVA with Šidák correction. *P*-values of <0.05 were considered statistically significant.

# Data Availability

All data are available in the published article and its online supplemental material.

# Supplementary Information

# Acknowledgements

We thank our colleagues and Dr. Masanori Iseki for helpful discussions, and Chinatsu Oyama, Miwa Tamura-Nakano and members in Communal Laboratory, RI, NCGM for technical assistance. This work was supported by JSPS KAKENHI Grant Numbers 20K07557 (M Tenno), 17K08802, 17KT0132, 22K07142, and 25K10406 (S Takaki), and The Grants for National Center for Global Health and Medicine (21A1023 and 24A1017) (S Takaki).

## Author Contributions

M Tenno: conceptualization, data curation, formal analysis, funding acquisition, validation, investigation, methodology, and writing—original draft.
S Takaki: conceptualization, resources, data curation, supervision, funding acquisition, validation, investigation, methodology, and writing—original draft, review, and editing.

## Conflict of Interest Statement

The authors declare that they have no conflict of interest.

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
