## [Reviewer comments · Life Science Alliance]

Lnk/Sh2b3 regulates initiation and severity of autoimmune insulinitis and contributes to diabetes risk

Mari Tenno and Satoshi Takaki

DOI: <https://doi.org/10.26508/lsa.202503332>

Corresponding author(s): Satoshi Takaki, National Center for Global Health and Medicine

Review Timeline:

Submission Date:	2025-03-30
Editorial Decision:	2025-04-28
Revision Received:	2025-08-14
Editorial Decision:	2025-09-09
Revision Received:	2025-09-17
Accepted:	2025-09-19

Scientific Editor: Tim Fessenden

Transaction Report:

April 28, 2025

Re: Life Science Alliance manuscript #LSA-2025-03332-T

Prof. Satoshi Takaki
National Center for Global Health and Medicine
Department of Immune Regulation
1-7-1 Kohnodai
Chiba 272-8516
Japan

Dear Dr. Takaki,

Thank you for submitting your manuscript entitled "Lnk/Sh2b3 regulates initiation and severity of autoimmune insulinitis and contributes to diabetes risk" to Life Science Alliance. The manuscript was assessed by expert reviewers, whose comments are appended to this letter.

As you will see, all reviewers commended the intriguing findings that link the adaptor Lnk/Sh2b3 with autoimmune activation involving dendritic cells and T cells. However, reviewers sought improved evidence and discussion of several points.

Namely, Reviewers 1 and 3 both remarked that the involvement of B cells, which contribute to T1 diabetes and are affected by CD40/CD40L signaling, was not examined in this work. While a full exploration of B cells is not required, the potential roles for B cells not addressed here must be discussed in a revised manuscript. Similarly, Reviewers 1 and 2 felt that claims on IL-27 signaling should be toned down. Next, Reviewers requested increased n for data in Figure 2. Reviewer 1 noted the flow cytometry staining for DC subsets should be refined to detect potential involvement of monocytes and monocyte-derived DC. Finally validation and greater details on mouse models were sought by Reviewers 1 and 2: by documenting numbers of pancreatic DC levels at baseline, describing phenotype and chimerism of MyD88 KO, and if possible by reporting beta cell loss over time across mouse models.

Thank you for this interesting contribution to Life Science Alliance. We are looking forward to receiving your revised manuscript.

Sincerely,

- A letter addressing the reviewers' comments point by point.
- An editable version of the final text (.DOC or .DOCX) is needed for copyediting (no PDFs).

B. MANUSCRIPT ORGANIZATION AND FORMATTING:

Reviewer #1 (Comments to the Authors (Required)):

In this study, Tenno and Takaki describe the impact of LNK/SH2B3 deficiency on diabetes development in a streptozotocin (STZ) model and the role of LNK/SH2B3 in regulating cytokine signaling in dendritic cells (DCs) and T cells. Overall, the authors demonstrate accelerated diabetes in LNK deficient mice treated with STZ and the mitigation of disease development by treating animals with blocking antibodies to CD40L or GM-CSF. They went on to characterize differences in the expression of surface receptors and downstream markers of stimulation on WT and LNK deficient DCs and T cells in the setting of STZ treatment. They extended this line of inquiry by ameliorating STZ-induced diabetes when LNK deficient mice were crossed to Rag2^{-/-} or MyD88^{-/-} lines. While the authors demonstrate a clear role for LNK in regulating autoimmunity in the setting of diabetes, the manuscript would be strengthened by the inclusion of cell types beyond DCs and T cells. Additionally, there's currently a gap between the data shown and the conclusions drawn that could be bridged by showing additional data or adjusting the scope of the conclusions.

Major points:

- The authors have in the past (Mori et al 2014) demonstrated baseline differences between WT and LNK deficient mice with increased splenic and lymph node DC populations. The authors should demonstrate that baseline levels of DC in pancreas are not elevated in LNK deficient mice.
- The sample size for several plots in Figure 2 are only n=2 at times which would not allow for meaningful statistical analysis as shown (e.g. TLR3 plot in Figure 2A includes n = 2 and n = 3). Given that the experiment was performed n=10 each genotype in Figure 1 and there are adequate numbers for most plots, please include if there are excluded values and increase the sample size to perform the proper statistical analysis.
- The authors conclude that the accumulation of activated CD8⁺ T cells in LNK deficient mice is due to IL-27-dependent signaling. They show that LNK deficient T cells exhibited increased IL-27R α expression at later time points and increased proportion of CD122⁺, Sca-1⁺, and T-bet⁺ cells. However, the selected activation markers aren't only upregulated by IL-27 signaling (eg. Sca-1 can be upregulated by TCR stimulation as well as IFN- γ stimulation). Evidence directly linking IL-27 with these markers, such as by stimulation with IL-27 or by blocking IL-27 signaling, could be included. Otherwise, conclusions should be adjusted.
- While the anti-CD40L treatment did result in decreased diabetes development in LNK deficient mice, this is not specific to DC/T-cell interactions as other cell types, such as B cells and monocytes/macrophages, express CD40 and are impacted by CD40L blockade. Please provide evidence about other cell types or adjust conclusion.
- The authors should show flow cytometry for the cDC1 and cDC2 populations in the supplemental data at minimum. It appears that the authors did look at CD11b⁺ and B220⁺, but hard to know if they looked at CD11b on DC populations. CD11b (negative on cDC1s, helps exclude cDC2s) could be included in the DC stain to look at these populations or use more specific XCR1 (highly specific for cDC1) or CD24 (high on cDC1s). In the NOD model, Tang's group showed a monocytic origin of CD11c⁺ cells in inflamed islets in that model not dependent on Zbtb46, while Unanue's group showed Batf3-dependent CD11b⁺CD103⁺ DCs are essential for disease initiation in NOD mice (PMID: 28550204). Thus, DC-like cells derived from monocytes could be present in the authors infiltrate. Please show the flow cytometry plots and discuss the possibility.
- The authors attribute MyD88 deficient mice not developing diabetes after low-dose STZ injection to DCs specifically. However, MyD88 is critical in TLR signaling for numerous cell types, including B and monocytes that would be reconstituted in the MyD88

deficient chimera model. MyD88 deficient mice are susceptible to infection and illness. Were any phenotypical differences in these mice, such as decreased weight, observed? Similarly, please show chimerism data for Supplemental Figure 3 in DC compartment for the tissue if making claim about radioresistant and sensitive celltype contribution to disease.

- The authors discuss insulin resistance in the introduction given their previous work, however, STZ-induced diabetes is a model of type 1 diabetes, not type 2 diabetes or insulin resistance. In a brief literature search, there are other groups with published work (<https://pubmed.ncbi.nlm.nih.gov/40048557/>) or abstracts on the topic (https://journals.aai.org/jimmunol/article/210/1_Supplement/77.03/265073/Lnk-Sh2b3-modulates-bioenergetic-metabolism-of) that should be cited as more relevant and further support the authors claim about role of LNK in type 1 diabetes. Otherwise, how adipose inflammation associated with T1D is not clear.

Minor points:

- Although there is a historic use of the term Lnk^{-/-}, the nomenclature including UCSC genome browser, The MGI database and the International Mouse Strain Resource have all referred to the gene as Sh2b3. When referring to the gene name, the authors should use the adopted nomenclature of Sh2b3 rather than Lnk for the gene name. If the authors would like to use LNK rather than SH2B3, either seems appropriate.
- Please keep the format for β cells consistent (currently a mix of " β cells" and " β -cells").
- Clarify non-specific language (eg. "about one week or so").
- The blood glucose curves would benefit from including n in the figure for ease of reading rather than just putting in figure legend.
- MFI should be defined.
- For the low-dose administration, a schematic of injection schedule would improve reader understanding, particularly when also administering blocking antibody to the mice.

Reviewer #2 (Comments to the Authors (Required)):

1. A short summary of the paper, including description of the advance offered to the field.

This work presents the role of Lnk/SH2B3 in destructive insulinitis utilizing a knockout mouse and streptozotocin-induced diabetes. Knockout mice were more susceptible to insulinitis, which was shown to be dependent on activated pancreatic dendritic cells through multiple confirmatory experimental methods, including antibody blockade and bone marrow chimeras. These efforts represent an advance in the understanding of the impact of SH2B3 on dendritic cell function in response to inflammatory insult to the pancreas, relevant to the pathogenesis of type 1 diabetes.

2. For each main point of the paper, please indicate if the data are strongly supportive. If not, explicitly state the additional experiments essential to support the claims made and the timeframe that these would require.

The data largely support the claims made throughout the manuscript. Some minor revisions to analysis and discussion of results would improve the support, as follows:

Investigation of insulinitis are shown as representative images. Quantitative analysis of the beta cell loss across all experimental animals would provide better support, particularly in Figure 4. This analysis would be expected to be completed within weeks and is possible using open-source software for histological analysis.

Flow cytometry plots should include negative controls (unstained samples or FMO controls) in the histograms to visually confirm gate placement for marker positivity analysis.

In Figure 2E, the ratio of co-inhibitory to co-stimulatory molecules is mentioned as important but is not calculated for the experimental data. This would be simple to perform and report within a week.

Finally, the supposition of IL-27 loops is conjecture but is stated as a more definitive result in the abstract. Suggest revision.

3. Lastly, indicate any additional issues you feel should be addressed (text changes, data presentation, statistics etc.).

The authors are advised to carefully review the manuscript to ensure that sentences referencing results from other papers are more clearly described as such throughout the manuscript (page 9, line 151 as an example).

In addition, SH2B3 has been knocked out in the NOD mouse and the impacts thoroughly described. This publication should be reviewed and compared to the current results, and included as a reference.

Figure 1 -

There are no visible error bars in panels A and C. The representative images are somewhat oversaturated.

Figure 2 -

The panels should be reordered in the figure so that B comes before C.

Discussion of Figure 2B needs further description of why the authors claim the results for CD40, CD80, and CD86 expression do not indicate general activation of dendritic cells

Figure 3 -

The discussion of IL-27 biology would benefit from some critical review and editing of the content.

Figure 4 -

In line 202-203, the authors note that anti-CD40L treatment blocked CD40-CD40L interaction seemingly as a summary result of the experiment, when this is just a statement of how antibody blockade works. Revision of this sentence is recommended.

Reviewer #3 (Comments to the Authors (Required)):

Tenno and Takaki hypothesize that the adaptor protein Lnk/Sh2b3, which negatively regulates cytokine signaling, drives development of type 1 diabetes (T1D) via cDC1-dependent mechanism. First, they assess diabetes susceptibility in knockout mice following low-dose treatment of streptozotocin (STZ), and show that the absence of Lnk drives hyperglycemia and insulinitis. Next, they show that these events are accompanied by pancreatic infiltration of CD8+ T-cells and activated cDC1s, and that lack of lymphocytes or MyD88 abolishes the high T1D susceptibility of Lnk^{-/-} mice. Lastly, they perform studies with cytokine neutralizing antibodies, and the data suggests that the aforementioned biology is driven by GM-CSF and IL-27 dependent mechanisms.

Overall, the premise of the manuscript is interesting and some of data looks promising, but additional experiments (discussed below) would significantly improve the quality of the manuscript and strengthen the story. In addition, certain aspects should be clarified or discussed further by the authors.

Major comments:

1) In addition to discussed effects on myeloid cells and T cells, Lnk has been shown to regulate normal B cell development, and islet-reactive B cells have been reported to play a crucial role in T1D pathogenesis (by presenting antigens to T cells and producing cytokines and autoantibodies). Moreover, rare Lnk coding variants in lupus patients were recently shown to impair B cell tolerance and predispose to autoimmunity (DOI: 10.1084/jem.20221080).

This raises the question whether B cells should be characterized in the current manuscript as well - what effects do the authors see in the B cell compartment (in secondary lymphoid organs and in the pancreas) in their Lnk^{-/-} mouse model? Do these B cells interact with T cells? This is especially relevant as authors claim on lines 118-129 that the results from Rag and BM chimeric mice indicate that beta-cell destruction was mediated by T cells (but similarly, B cells would also be affected in these model systems).

2) Text on lines 62-63 mentions that Lnk missense mutation (R262W) is a risk variant for various autoimmune diseases, including T1D. What is known about the effect of this mutation on function - is it leading to gain or loss of function, is the mutation occurring in a critical structural site of the protein that has been shown to interact with its binding partners? Additional discussion on this topic could benefit the premise of the manuscript.

3) Lines 77-78 state that "relationship between the function of Lnk pathological conditions in T1D development has yet to be fully elucidated." This is not entirely correct, as Watson et al (Diabetes, 2025; <https://doi.org/10.2337/db24-0655>) recently published a similar study on Lnk and its effect on T-cell driven pathogenesis of T1D. The authors are encouraged to reference this article in their manuscript and discuss the main similarities and differences of their findings.

4) As NOD mice are traditionally used for preclinical studies for T1D biology, it would be important that the authors clarify how STZ is hypothesized to induce T cell driven insulinitis. Which immune signaling pathways and cytokines are induced by STZ? Would similar pathology-inducing events happen in NOD mice or in human T1D? These points would be important to clarify to understand the translational relevance of the current study and whether the described dependence of T1D development on GM-CSF and IL-27 would also hold true in a more spontaneous setting.

5) In multiple Figures, the authors make quite strong claims from studies with limited number of mice (e.g. multiple subpanels in Figure 2 and 3 with N=2-3 mice per group). For sufficient conclusions, the authors are requested to repeat some of those studies with a larger N to validate that the biological effect is reproducible.

6) For overall clarity of the manuscript, the text should be reformatted into separate "Results" and "Discussion" sections. For example, a lot of the text on lines 157-169 could be condensed or moved to discussion. The main body of the manuscript should be focused mainly on the results of the present study. The authors are encouraged to clearly separate statements based on literature (using a present tense or by using phrasing such as "has been shown" or "Author x et al have reported") from statements based on data from the current study (using past tense, e.g. "IL-27 was upregulated in Lnk^{-/-} cDC1s following STZ treatment"). In its current state, it is difficult to decipher what are the novel findings in this manuscript.

Minor comments:

1) When first introducing Lnk, mention what words the acronym consists of (lymphocyte adapter protein)

2) Line 76 has a typo (INF instead of IFN)

- 3) Lines 113-114: Please include the numbers of B220+, CD11b+, and F4/80+ as a supplementary Figure.
- 4) Lines 114-115: Is there an increase in the number (not frequency) of CD8+ T cells in Lnk^{-/-} pancreas and draining lymph nodes at steady state?
- 5) Please provide additional clarification on how the statistical analysis was performed in Figures 1A, 4A, 5A and in supplementary figures S2 A-B, S3 A-B. Was the significance calculated as function of time (days) and taking into account repeated measures?
- 6) Figure 2: Panel C is currently placed before panel B in the figure. Also, please add an illustration/schematic in the figure explaining the treatment regimen (when was STZ and anti-CD40L antibody administered etc)

Point-by-point response to Reviewers' comments:**Reviewer #1**

In this study, Tenno and Takaki describe the impact of LNK/SH2B3 deficiency on diabetes development in a streptozotocin (STZ) model and the role of LNK/SH2B3 in regulating cytokine signaling in dendritic cells (DCs) and T cells. Overall, the authors demonstrate accelerated diabetes in LNK deficient mice treated with STZ and the mitigation of disease development by treating animals with blocking antibodies to CD40L or GM-CSF. They went on to characterize differences in the expression of surface receptors and downstream markers of stimulation on WT and LNK deficient DCs and T cells in the setting of STZ treatment. They extended this line of inquiry by ameliorating STZ-induced diabetes when LNK deficient mice were crossed to Rag2^{-/-} or MyD88^{-/-} lines. While the authors demonstrate a clear role for LNK in regulating autoimmunity in the setting of diabetes, the manuscript would be strengthened by the inclusion of cell types beyond DCs and T cells. Additionally, there's currently a gap between the data shown and the conclusions drawn that could be bridged by showing additional data or adjusting the scope of the conclusions.

Major points:

- *The authors have in the past (Mori et al 2014) demonstrated baseline differences between WT and LNK deficient mice with increased splenic and lymph node DC populations. The authors should demonstrate that baseline levels of DC in pancreas are not elevated in LNK deficient mice.*

>> First of all, we really thank the reviewer for giving us positive comments. We added a graph showing the cell numbers of DC population in the steady state pancreas in Supplement S1D. There was no difference between WT and Lnk KO in both proportions and absolute cell numbers of cDC1 and cDC2 cells in the pancreas.

- *The sample size for several plots in Figure 2 are only n=2 at times which would not allow for meaningful statistical analysis as shown (e.g. TLR3 plot in Figure 2A includes n = 2 and n = 3). Given that the experiment was performed n=10 each genotype in Figure 1 and there are adequate numbers for most plots, please include if there are excluded values and increase the sample size to perform the proper statistical analysis.*

>> We added data and increased the sample size in Fig 2A, 2C, 2D, and reordered some graphs showing related parameters. Since WT showed little inflammatory response with a small amount of STZ, and the data variation was slight, n = 4 was considered sufficient.

- *The authors conclude that the accumulation of activated CD8⁺ T cells in LNK deficient mice is due to IL-27-dependent signaling. They show that LNK deficient T cells exhibited increased IL-27R α expression at later time points and increased proportion of CD122⁺, Sca-1⁺, and T-bet⁺ cells. However, the selected activation markers aren't only upregulated by IL-27 signaling (eg. Sca-1 can be upregulated by TCR stimulation as well as IFN- γ stimulation). Evidence directly linking IL-27 with these markers, such as by stimulation with IL-27 or by blocking IL-27 signaling, could be included. Otherwise, conclusions should be adjusted.*

>> We revised the description on IL-27 and IL-27R in Discussion section, p15 line 246-249. "Thus, it seems that IL-27-dependent signaling contributes in part to the accumulation of CD8⁺ effector T-cells in the pancreas without Lnk. Further experiments, however, will be required to confirm if IL-27 plays critical roles in developing diabetes after β cell injury in the absence of Lnk."

We also cited the work by Watson et al (Diabetes, 2025) and added the discussion on possible functions of Lnk in developing type 1 diabetes in Discussion section, p16-17 line 272-2848. Lnk-deficiency potently promotes pathogenic CD8⁺ T cells with enhanced survival and effector function by enhanced activation of the JAK/STAT pathways via IL-2, IL-15 and IFN γ .

• *While the anti-CD40L treatment did result in decreased diabetes development in LNK deficient mice, this is not specific to DC/T-cell interactions as other cell types, such as B cells and monocytes/macrophages, express CD40 and are impacted by CD40L blockade. Please provide evidence about other cell types or adjust conclusion.*

>> We really thank the reviewer for the comment. Accordingly, we performed additional experiments and examined the involvement of B cells by depleting B cells using anti-CD20. Repeated injection of anti-CD20 efficiently depleted B cells in Lnk^{-/-} or WT type mice within 7 days. Anti-CD20-treated Lnk^{-/-} mice manifested hyperglycemia after STZ injection, comparable to those treated with control IgG antibodies. WT mice did not show hyperglycemia despite anti-CD20 treatment. We added the results and presented them in the new Supplemental Figure S5. In addition, the percentage and absolute number of B220⁺, CD11b, and F4/80⁺ cells remained unchanged on day 7 after STZ administration as shown in Fig. S2. The MFI of CD40 in these cells also remained unchanged. GM-CSF neutralization, which has little effect on B cells, protected Lnk^{-/-} mice from insulinitis and diabetes induced by low-dose STZ injection. These data further implied that anti-CD40L-treatment blocked DC/T-cell interaction, but not B/T-cell interaction via CD40–CD40L. We added the description in Discussion section, p15-16 line 250-259, and Result section p9-10 line 153-155.

• *The authors should show flow cytometry for the cDC1 and cDC2 populations in the supplemental data at minimum. It appears that the authors did look at CD11b⁺ and B220⁺, but hard to know if they looked at CD11b on DC populations. CD11b (negative on cDC1s, helps exclude cDC2s) could be included in the DC stain to look at these populations or use more specific XCR1 (highly specific for cDC1) or CD24 (high on cDC1s). In the NOD model, Tang's group showed a monocytic origin of CD11c⁺ cells in inflamed islets in that model not dependent on Zbtb46, while Unanue's group showed Batf3-dependent CD11b⁺CD103⁺ DCs are essential for disease initiation in NOD mice (PMID: 28550204). Thus, DC-like cells derived from monocytes could be present in the authors infiltrate. Please show the flow cytometry plots and discuss the possibility.*

>> We added the flow cytometry plots and gating strategies in the revised Fig. S2A. There were no changes in the number of CD11b⁺F4/80⁻ (monocytes) or CD11b⁺F4/80⁺ (macrophages) cells. We gated on CD45⁺F4/80⁻CD64⁻CD11c⁺MHCII⁺ cells to exclude macrophages and monocytes, and further divided into CD103⁺ cDC1 and CD11b⁺ cDC2. as shown in Fig. S1 D (steady state) and Fig. 1 F (after STZ administration). We focused on CD103b⁺ cDC1 in further analysis, since there were no significant changes in CD11b⁺ cDC2 as well as in the CD11b⁺F4/80⁻ monocyte-derived cells and CD11b⁺F4/80⁺ macrophages.

• *The authors attribute MyD88 deficient mice not developing diabetes after low-dose STZ injection to DCs specifically. However, MyD88 is critical in TLR signaling for numerous cell types, including B and monocytes that would be reconstituted in the MyD88 deficient chimera model. MyD88 deficient mice are susceptible to infection and illness. Were any phenotypical differences in these mice, such as decreased weight, observed? Similarly, please show chimerism data for Supplemental Figure 3 in DC compartment for the tissue if making claim about radioresistent and sensitive celltype contribution to disease.*

>> We appreciate the reviewer for the comment. Accordingly, we performed additional experiments and examined the involvement of B cells by depleting B cells using anti-CD20. Repeated injection of anti-CD20 efficiently depleted B cells in Lnk^{-/-} or WT type mice within 7 days. Anti-CD20-treated Lnk^{-/-} mice manifested hyperglycemia after STZ injection, comparable to those treated with control IgG antibodies. WT mice did not show hyperglycemia despite anti-CD20 treatment. We added the results and presented them in the new Supplemental Figure S5. These data suggested that MyD88-dependent activation of non-B cells is critical for developing diabetic conditions after

low-dose STZ injection in Lnk^{-/-} mice. We added the description in Discussion section, p15-16 line 250-259 We added flowcytometry plots showing chimerism in reconstituted DC fraction in pancreas after bone marrow transfer in Supplemental Figure S4. Most DC cells were transferred donor BM-derived cells. Myd88/Lnk double deficient mice did not show any phenotypical difference compared to MyD88 deficient mice in the steady state conditions.

- *The authors discuss insulin resistance in the introduction given their previous work, however, STZ-induced diabetes is a model of type 1 diabetes, not type 2 diabetes or insulin resistance. In a brief literature search, there are other groups with published work (<https://pubmed.ncbi.nlm.nih.gov/40048557/>) or abstracts on the topic (https://journals.aai.org/jimmunol/article/210/1_Supplement/77.03/265073/Lnk-Sh2b3-modulates-bioenergetic-metabolism-of) that should be cited as more relevant and further support the authors claim about role of LNK in type 1 diabetes. Otherwise, how adipose inflammation associated with T1D is not clear.*

>> We thank the reviewer for the comment. We cited the work by Watson et al (Diabetes, 2025; <https://doi.org/10.2337/db24-0655>, <https://pubmed.ncbi.nlm.nih.gov/40048557/>) and added the discussion on possible functions of Lnk in developing type 1 diabetes in Discussion section, p16-17 line 272-284. We also carefully rewrote and introduced our previous observation on adipose tissue inflammation leading to mild glucose intolerance in Lnk^{-/-} mice, and added description in Result section “When Lnk^{-/-} mice were fed a normal chow diet, blood glucose levels were slightly elevated (Fig. S1 A) due to adipose inflammation as previously reported”, p6 line 93.

Minor points:

- *Although there is a historic use of the term Lnk^{-/-}, the nomenclature including UCSC genome browser, The MGI database and the International Mouse Strain Resource have all referred to the gene as Sh2b3. When referring to the gene name, the authors should use the adopted nomenclature of Sh2b3 rather than Lnk for the gene name. If the authors would like to use LNK rather than SH2B3, either seems appropriate.*

>> We thank for the comment. We would like to use Lnk in our manuscript, as our continuous contribution from gene cloning, establishment of the KO mice to functional analysis.

- *Please keep the format for β cells consistent (currently a mix of " β cells" and " β -cells").*

>> We really appreciate the reviewer’s comment. We corrected " β -cells" to " β cells" and keep the format for β cells consistent.

- *Clarify non-specific language (eg. "about one week or so").*

>> We corrected and clarified non-specific languages, p6 line 100.

- *The blood glucose curves would benefit from including n in the figure for ease of reading rather than just putting in figure legend.*

>> We thank for the comment. We revised blood glucose curves in Fig. 1A, 4A, 5A, S3, S4 and the legends, and included “n” in the figures.

- *MFI should be defined.*

>> We added mean fluorescence intensity (MFI) in the first description of MFI in p34 line 631.

- *For the low-dose administration, a schematic of injection schedule would improve reader understanding, particularly when also administering blocking antibody to the mice.*

>> We appreciate the reviewer for the comment. We added schematics of injection schedule of STZ and antibodies in Fig. 1A, 4A and 5A.

Reviewer #2

1. A short summary of the paper, including description of the advance offered to the field.

This work presents the role of Lnk/SH2B3 in destructive insulinitis utilizing a knockout mouse and streptozotocin-induced diabetes. Knockout mice were more susceptible to insulinitis, which was shown to be dependent on activated pancreatic dendritic cells through multiple confirmatory experimental methods, including antibody blockade and bone marrow chimeras. These efforts represent an advance in the understanding of the impact of SH2B3 on dendritic cell function in response to inflammatory insult to the pancreas, relevant to the pathogenesis of type 1 diabetes.

>> We thank the reviewer for giving us positive comments.

2. For each main point of the paper, please indicate if the data are strongly supportive. If not, explicitly state the additional experiments essential to support the claims made and the timeframe that these would require.

The data largely support the claims made throughout the manuscript. Some minor revisions to analysis and discussion of results would improve the support, as follows:

Investigation of insulinitis are shown as representative images. Quantitative analysis of the beta cell loss across all experimental animals would provide better support, particularly in Figure 4. This analysis would be expected to be completed within weeks and is possible using open-source software for histological analysis.

>> We appreciate the comment. Accordingly, we performed quantitative analysis of the β cell loss in all histology figures, and the results were presented as graphs in Fig. 1C, 4B and 5B, Supplemental Fig. S1B, S3A, S3B S4A, and S4B.

Flow cytometry plots should in the histograms to visually confirm gate placement for marker positivity analysis.

>> We revised histograms and included unstained or FMO controls in Fig. 3C, 3D and 3E.

In Figure 2E, the ratio of co-inhibitory to co-stimulatory molecules is mentioned as important but is not calculated for the experimental data. This would be simple to perform and report within a week.

>> We thank for the comment. We calculated the Lnk/wt ratio in MFI of co-inhibitory and co-stimulatory molecules expressed on Lnk^{-/-} compared to WT cells, and showed the ratio below each histogram in Fig. 2B and 2E. Most (3 out of 4) of co-stimulatory molecules were up-regulated on Lnk^{-/-} cells, whereas only PD-L1 (1 out of 3) was marginally up-regulated on Lnk^{-/-} cells compared with WT cells.

Finally, the supposition of IL-27 loops is conjecture but is stated as a more definitive result in the abstract. Suggest revision.

>> We revised the description on IL-27 and IL-27R in Discussion section, p15 line 246-249. "Thus, it seems that IL-27-dependent signaling contributes in part to the accumulation of CD8⁺ effector T-cells in the pancreas without Lnk. Further experiments, however, will be required to confirm if IL-27 plays critical roles in developing diabetes after β cell injury in the absence of Lnk." We also revised the descriptions in abstract p2 line 30 and conclusion section p15 line 287.

3. Lastly, indicate any additional issues you feel should be addressed (text changes, data presentation, statistics etc.).

The authors are advised to carefully review the manuscript to ensure that sentences referencing results from other papers are more clearly described as such throughout the manuscript (page 9, line 151 as an example).

>> We really appreciate the reviewer for the suggestion. Accordingly, we revised and separated sentences referencing results into “Result” and descriptions on other papers into “Discussion” sections.

In addition, SH2B3 has been knocked out in the NOD mouse and the impacts thoroughly described. This publication should be reviewed and compared to the current results, and included as a reference.

>> We thank the reviewer for the comment. We cited the work by Watson et al (Diabetes, 2025; <https://doi.org/10.2337/db24-0655>, <https://pubmed.ncbi.nlm.nih.gov/40048557/>) and added the discussion on possible functions of Lnk in developing type 1 diabetes in Discussion section, p16-17 line 272-284. Lnk-deficiency exacerbated diabetes in multiple mouse models, including NOD. Lnk-deficiency potently promotes pathogenic CD8+ T cells with enhanced survival and effector function by enhanced activation of the JAK/STAT pathways via IL-2, IL-15 and IFN γ .

Figure 1 -

There are no visible error bars in panels A and C. The representative images are somewhat oversaturated.

>> WT showed little inflammatory response with a small amount of STZ, and the data variation was slight in WT, and error bars were smaller than symbols in WT. Concerning images showing insulinitis, we performed quantitative analysis of the β cell loss in all histology figures, and the results were presented as graphs in Fig. 1C, 4B and 5B, Supplemental Fig. S1B, S3A, S3B S4A, and S4B.

Figure 2 -

The panels should be reordered in the figure so that B comes before C.

Discussion of Figure 2B needs further description of why the authors claim the results for CD40, CD80, and CD86 expression do not indicate general activation of dendritic cells

>> We thank for the comment. We reordered some graphs showing related parameters, and revised Fig. 2. We calculated the Lnk/wt ratio in MFI of co-inhibitory and co-stimulatory molecules expressed on Lnk $^{-/-}$ compared to WT cells, and showed the ratio below each histogram in Fig. 2B and 2E. Most (3 out of 4) of co-stimulatory molecules were up-regulated on Lnk $^{-/-}$ cells, whereas only PD-L1 (1 out of 3) was marginally up-regulated on Lnk $^{-/-}$ cells compared with WT cells. We added the description in Discussion section, p14 line 225-236.

Figure 3 -

The discussion of IL-27 biology would benefit from some critical review and editing of the content.

>> We added a review by Meka et al., p13 line 214-218, “IL-27 is a unique cytokine with reported immunostimulatory and immunosuppressive effects that are exerted through STAT1/3-dependent cascades and thought to be involved in various autoimmune diseases including T1D (Meka et al., 2015).” We also revised the description on IL-27 and IL-27R in Discussion section, p15 line 246-249, “Thus, it seems that IL-27-dependent signaling contributes in part to the accumulation of CD8+ effector T-cells in the pancreas without Lnk. Further experiments, however, will be required to confirm if IL-27 plays critical roles in developing diabetes after β cell injury in the absence of Lnk.”

Figure 4 -

In line 202-203, the authors note that anti-CD40L treatment blocked CD40-CD40L interaction seemingly as a summary result of the experiment, when this is just a statement of how antibody blockade works. Revision of this sentence is recommended.

>> We really thank the reviewer for the comment. Accordingly, we performed additional experiments and examined the involvement of B cells by depleting B cells using anti-CD20. Repeated injection of anti-CD20 efficiently depleted B cells in Lnk $^{-/-}$ or WT type mice within 7 days. Anti-CD20-treated Lnk $^{-/-}$ mice manifested hyperglycemia after STZ injection, comparable to those

treated with control IgG antibodies. WT mice did not show hyperglycemia despite anti-CD20 treatment. We added the results and presented them in the new Supplemental Figure S5. In addition, the percentage and absolute number of B220+, CD11b, and F4/80+ cells remained unchanged on day 7 after STZ administration as shown in Fig. S2. The MFI of CD40 in these cells also remained unchanged. GM-CSF neutralization, which has little effect on B cells, protected Lnk^{-/-} mice from insulinitis and diabetes induced by low-dose STZ injection. These data further implied that anti-CD40L-treatment blocked DC/T-cell interaction, but not B/T-cell interaction via CD40-CD40L. We added the description in Discussion section, p15-16 line 250-259, and Result section p10 line 154-155.

Reviewer #3

Tenno and Takaki hypothesize that the adaptor protein Lnk/Sh2b3, which negatively regulates cytokine signaling, drives development of type 1 diabetes (T1D) via cDC1-dependent mechanism. First, they assess diabetes susceptibility in knockout mice following low-dose treatment of streptozotocin (STZ), and show that the absence of Lnk drives hyperglycemia and insulinitis. Next, they show that these events are accompanied by pancreatic infiltration of CD8+ T-cells and activated cDC1s, and that lack of lymphocytes or MyD88 abolishes the high T1D susceptibility of Lnk^{-/-} mice. Lastly, they perform studies with cytokine neutralizing antibodies, and the data suggests that the aforementioned biology is driven by GMCSF and IL-27 dependent mechanisms.

Overall, the premise of the manuscript is interesting and some of data looks promising, but additional experiments (discussed below) would significantly improve the quality of the manuscript and strengthen the story. In addition, certain aspects should be clarified or discussed further by the authors.

>> We thank the reviewer for giving us positive comments.

Major comments:

1) In addition to discussed effects on myeloid cells and T cells, Lnk has been shown to regulate normal B cell development, and islet-reactive B cells have been reported to play a crucial role in T1D pathogenesis (by presenting antigens to T cells and producing cytokines and autoantibodies). Moreover, rare Lnk coding variants in lupus patients were recently shown to impair B cell tolerance and predispose to autoimmunity (DOI: 10.1084/jem.20221080).

This raises the question whether B cells should be characterized in the current manuscript as well - what effects do the authors see in the B cell compartment (in secondary lymphoid organs and in the pancreas) in their Lnk^{-/-} mouse model? Do these B cells interact with T cells? This is especially relevant as authors claim on lines 118-129 that the results from Rag and BM chimeric mice indicate that beta-cell destruction was mediated by T cells (but similarly, B cells would also be affected in these model systems).

>> We really thank the reviewer for the comment. Accordingly, we performed additional experiments and examined the involvement of B cells by depleting B cells using anti-CD20. Repeated injection of anti-CD20 efficiently depleted B cells in Lnk^{-/-} or WT type mice within 7 days. Anti-CD20-treated Lnk^{-/-} mice manifested hyperglycemia after STZ injection, comparable to those treated with control IgG antibodies. WT mice did not show hyperglycemia despite anti-CD20 treatment. We added the results and presented them in the new Supplemental Figure S5. In addition, the percentage and absolute number of B220+, CD11b, and F4/80+ cells remained unchanged on day 7 after STZ administration as shown in Fig. S2. The MFI of CD40 in these cells also remained

unchanged. GM-CSF neutralization, which has little effect on B cells, protected Lnk^{-/-} mice from insulinitis and diabetes induced by low-dose STZ injection. These data further implied that anti-CD40L-treatment blocked DC/T-cell interaction, but not B/T-cell interaction via CD40-CD40L. We added the description in Discussion section, p15-16 line 250-259, and Result section p10 line 154-155.

2) Text on lines 62-63 mentions that Lnk missense mutation (R262W) is a risk variant for various autoimmune diseases, including T1D. What is known about the effect of this mutation on function - is it leading to gain or loss of function, is the mutation occurring in a critical structural site of the protein that has been shown to interact with its binding partners? Additional discussion on this topic could benefit the premise of the manuscript.

>> We thank the reviewer for the comment. We cited the works by Watson et al (Diabetes, 2025) and by Wang et al (Circ Res, 2016), which show the R262W risk variant strongly associates with T1D encodes a reduced function (hypomorphic) protein, and added the discussion on possible functions of Lnk in developing type 1 diabetes in Discussion section, p16-17 line 272-284. Lnk-deficiency exacerbated diabetes in multiple mouse models, including NOD. Lnk-deficiency potently promotes pathogenic CD8⁺ T cells with enhanced survival and effector function by enhanced activation of the JAK/STAT pathways via IL-2, IL-15 and IFN γ .

3) Lines 77-78 state that "relationship between the function of Lnk pathological conditions in T1D development has yet to be fully elucidated." This is not entirely correct, as Watson et al (Diabetes, 2025; <https://doi.org/10.2337/db24-0655>) recently published a similar study on Lnk and its effect on T-cell driven pathogenesis of T1D. The authors are encouraged to reference this article in their manuscript and discuss the main similarities and differences of their findings.

>> We thank the reviewer for the comment. We cited the work by Watson et al (Diabetes, 2025; <https://doi.org/10.2337/db24-0655>) and added the discussion on possible functions of Lnk in developing type 1 diabetes in Discussion section, p16-17 line 272-276. We also added the discussion on the potential use of JAK inhibitors in T1D, p17 line 276-284. JAK inhibitors have shown efficacy in preserving β -cell function in the NOD mouse model (Trivedi et al., 2017) and in human T1D subjects (Waibel et al., 2023). Our findings added regulatory functions of Lnk in early stages after initial β -cell injury in developing autoimmune diabetes. We revealed enhanced activation of cDC1 and expansion of CD8⁺ T cells without Lnk by potentially mediated through GM-CSF and possibly IL-27, both of which also activate JAK/STAT pathways. Thus, it is promising that JAK inhibitors will efficiently prevent or attenuate β -cell destruction and progression of diabetes, accompanied by reduced functions of Lnk with the risk variants of LNK/SH2B3 gene (Wang et al., 2016) for T1D.

4) *As NOD mice are traditionally used for preclinical studies for T1D biology, it would be important that the authors clarify how STZ is hypothesized to induce T cell driven insulinitis. Which immune signaling pathways and cytokines are induced by STZ? Would similar pathology-inducing events happen in NOD mice or in human T1D? These points would be important to clarify to understand the translational relevance of the current study and whether the described dependence of T1D development on GMCSF and IL-27 would also hold true in a more spontaneous setting.*

>> MLD-STZ damages (stresses) pancreatic β cells, but the damage is minimal, so the subsequent infiltration of immune cells is the primary cause of cell-dependent T1D. It has also been reported that β cell damage due to (viral) infection proceed to human T1D. The importance of IL-27, particularly in T cells, has also been shown in the NOD mice. In this paper, the source of IL-27 is shown to be myeloid cells, but the detailed subsets and mechanisms are still unknown. In addition to JAK inhibitors as discussed in p17 line 276-284, GM-CSF blockade might become beneficial to prevent or attenuate β -cell destruction and progression of diabetes after infection in high-risk group for T1D.

5) In multiple Figures, the authors make quite strong claims from studies with limited number of mice (e.g. multiple subpanels in Figure 2 and 3 with N=2-3 mice per group). For sufficient conclusions, the authors are requested to repeat some of those studies with a larger N to validate that the biological effect is reproducible.

>> We added data and increased the sample size in Fig 2A, 2C, 2D, and reordered some graphs showing related parameters. Since WT showed little inflammatory response with a small amount of STZ, and the data variation was slight, n = 4 was considered sufficient.

6) For overall clarity of the manuscript, the text should be reformatted into separate "Results" and "Discussion" sections. For example, a lot of the text on lines 157-169 could be condensed or moved to discussion. The main body of the manuscript should be focused mainly on the results of the present study. The authors are encouraged to clearly separate statements based on literature (using a present tense or by using phrasing such as "has been shown" or "Author x et al have reported") from statements based on data from the current study (using past tense, e.g. "IL-27 was upregulated in Lnk^{-/-} cDC1s following STZ treatment"). In its current state, it is difficult to decipher what are the novel findings in this manuscript.

>> We really appreciate the reviewer's comment and suggestion. We reformatted our manuscript in the revised version into separate "Results" and "Discussion" sections.

Minor comments:

1) When first introducing Lnk, mention what words the acronym consists of (lymphocyte adapter protein)

>> We thank for the comment. Historically, the first paper on the cloning of Lnk gene (coding only partial C-terminal half of Lnk protein) by Huang and Hayashi et al (PNAS, 1995) did not mention the acronym for Lnk, unfortunately. We would like to use Lnk in our manuscript, as our continuous contribution from gene cloning (for the full-length protein), establishment of the KO mice to functional analysis.

2) Line 76 has a typo (INF instead of IFN)

>> Thank you very much for the comment. We corrected "INF" to "IFN".

3) Lines 113-114: Please include the numbers of B220⁺, CD11b⁺, and F4/80⁺ as a supplementary Figure.

>> We added the flow cytometry plots and gating strategies in the revised Fig. S2. There were no changes in the number of CD11b⁺F4/80⁻ (monocytes) or CD11b⁺F4/80⁺ (macrophages) cells. We gated on CD45⁺F4/80⁻CD64⁻CD11c⁺MHCII⁺ cells to exclude macrophages and monocytes, and further divided into CD103⁺ cDC1 and CD11b⁺ cDC2. as shown in Fig. S1D (steady state) and Fig. 1F. We focused on CD103⁺ cDC1 in further analysis, since there were no significant changes in CD11b⁺ cDC2 as well as in the CD11b⁺F4/80⁻ monocyte-derived cells and CD11b⁺F4/80⁺ macrophages.

4) Lines 114-115: Is there an increase in the number (not frequency) of CD8⁺ T cells in Lnk^{-/-} pancreas and draining lymph nodes at steady state?

>> The number of CD8⁺ T cells in Lnk^{-/-} pancreas and draining lymph nodes at steady state were added to Fig. S1 C and E along with FACS gate. The number of CD8⁺ T-cell in dLN in Lnk mice was comparable to WT in steady state.

5) Please provide additional clarification on how the statistical analysis was performed in Figures 1A,

4A, 5A and in supplementary figures S2 A-B, S3 A-B. Was the significance calculated as function of time (days) and taking into account repeated measures?

>> We thank for the comment. We provide how the statistical analysis was performed in each figure legend. One-way or two-way ANOVA with Šidák correction was used for comparing differences among two or more groups of data (Figs 1A, 4A, 5A, S2AB, and S3AB).

6) Figure 2: Panel C is currently placed before panel B in the figure. Also, please add an illustration/schematic in the figure explaining the treatment regimen (when was STZ and anti-CD40L antibody administered etc)

>> We thank for the comment. We reordered some graphs showing related parameters, and revised Fig. 2. We added schematics of injection schedule in Fig. 1A, Fig. 4A and Fig. 5A

September 9, 2025

RE: Life Science Alliance Manuscript #LSA-2025-03332-TR

Dr. Mari Tenno
Tokyo University of Science
Division of Cancer Cell Biology, Research Institute for Biomedical Sciences (RIBS)
Noda, Chiba
Japan

Dear Dr. Tenno,

Thank you for submitting your revised manuscript entitled "Lnk/Sh2b3 regulates initiation and severity of autoimmune insulinitis and contributes to diabetes risk". As you will see, all reviewers are now satisfied and recommend publication.

Reviewer 1 remarked on a discrepancy in glucose levels measured in relation to Fig S5. Please address this concern by amending the text appropriately or providing data to confirm disease status, although generating new data to resolve this issue is not required. Reviewer 2 made several minor suggestions to improve the text, which we invite you to consider. We would be happy to publish your paper in Life Science Alliance pending these changes and final revisions necessary to meet our formatting guidelines.

- Please add ORCID ID for corresponding author--you should have received instructions on how to do so.
- Please add the X and Bluesky handles of your host institute/organization, as well as your own and/or one of the authors in our system.
- The "Data Availability" section should be placed after the Materials & Methods section. Please consult our guidelines at <https://www.life-science-alliance.org/manuscript-prep#format>
- Please add your main and supplementary figure legends to the main manuscript text after the references section.
- Please add callouts for Figures 3B and S5A-B to your main manuscript text
- Please indicate the size of scale bars in the legends for Figures 1B, 4B, 5B, and S1B.
- LSA requires that all observations reported are supported with data, therefore please remove or amend the statement on line 100 for which data is not shown.

A. FINAL FILES:

B. MANUSCRIPT ORGANIZATION AND FORMATTING:

Thank you for your attention to these final processing requirements. Please revise and format the manuscript and upload materials as soon as you are able.

Sincerely,

Reviewer #1 (Comments to the Authors (Required)):

In this resubmission, Tenno and Takaki describe the impact of LNK/SH2B3 deficiency on diabetes development in a streptozotocin (STZ) model and the role of LNK/SH2B3 in regulating cytokine signaling in dendritic cells (DCs) and T cells. Overall, the major comments were addressed, and the authors present a much-improved manuscript that will provide advances to the field. The only criticism includes that the B cell depletion experiment presented in Supplemental Fig S5 does not demonstrate the same level of hyperglycemia results as shown in Figure 1 or the other experiments which clearly showed >300-400 mg/dl glucose measurements post-STZ. Both the control Ig and anti-CD20 treated Sh2b3^{-/-} mice in the S5 figure were only ~200mg/dL post-STZ. In the literature, this is not actually considered hyperglycemic. Do the authors have evidence that these experiments worked with other metrics of disease? If not, I would suggest caution about emphasizing the lack of effect of B cells. The authors have fully addressed all other comments.

Reviewer #2 (Comments to the Authors (Required)):

This work provides a comprehensive evaluation of the impacts of Lnk knockout on the T cell mediated destruction of islets in the STZ model of autoimmune diabetes. Knockout mice were more susceptible to insulinitis, with special emphasis on determining the role of dendritic cells (cDC1) in the pancreas, GM-CSF, and IL-27 in the initiation and perpetuation of T cell expansion and autoimmune attack. This effort provides additional knowledge when compared to the recent publication of Lnk knockout on the NOD mouse background, particularly in the context of dendritic cells. The claims made by the manuscript are well supported by the data provided and the revisions that have been made. However, the claim that "signaling is affected by the absence of Lnk" should be made more specific to the pathways included in analysis.

Specific issues to address prior to publication:

Pg 9 line 151 - needs reference to NOD observation

Pg 12 line 202 - pancreases pancreata

Pg 14 Line 220-224 - no references are given for these statements

Pg 15 line 252-3 - wild type mice did not show hyperglycemia after anti-CD20 treatment

Pg 15 line 253 - clarify statement on GM-CSF neutralization has little effect on B cells with evidence or reference

Pg 17 line 273-279 - Some suggested edits, but this addition is also somewhat redundant/repetitive with the final paragraph.

Suggest merging the two paragraphs and editing if the discussion of JAKi is desired.

"Our findings support regulatory functions of Lnk in response to beta cell injury in the STZ model of autoimmune diabetes. We revealed enhanced activation of cDC1 and expansion of CD8+ T cells in the absence of Lnk, potentially mediated through GM-CSF and IL-27, both of which also activate JAK/STAT pathways. Overall, these data provide further support for the use of JAK inhibitors in T1D"

Pg 33 line 598 - typo in "statistical"

Figures

Error bars need to be shown consistently, they can be rendered in gray instead of black, or shaded, to avoid obscuring data points

Figure references:

Figure S2 appears in the manuscript before Figure S1.

Only Figure S3A is referred to.

Reviewer #3 (Comments to the Authors (Required)):

The authors answered all my questions sufficiently and performed a significant number of new experiments and analyses that improved both quality and clarity of the manuscript.

September 19, 2025

RE: Life Science Alliance Manuscript #LSA-2025-03332-TRR

Dr. Satoshi Takaki
National Center for Global Health and Medicine
Department of Immune Regulation
1-7-1 Kohnodai, Ichikawa
Chiba 272-8516
Japan

Dear Dr. Takaki,

Thank you for submitting your Research Article entitled "Lnk/Sh2b3 regulates initiation and severity of autoimmune insulinitis and contributes to diabetes risk". It is a pleasure to let you know that your manuscript is now accepted for publication in Life Science Alliance. Congratulations on this interesting work.

As you evaluate proofs and copyediting and undertake any final changes, kindly verify that the y-axis for blood glucose shown in Fig S5A (mg/ml) is correct, in contrast to the y-axes showing mg/dl in other figure panels.

DISTRIBUTION OF MATERIALS:

Again, congratulations on a very nice paper. I hope you found the review process to be constructive and are pleased with how the manuscript was handled editorially. We look forward to future exciting submissions from your lab.

Sincerely,
